# Look beyond the Mirror: Laparoscopic Cholecystectomy in Situs Inversus Totalis—A Systematic Review and Meta-Analysis (and Report of New Technique)

**DOI:** 10.3390/diagnostics12051265

**Published:** 2022-05-19

**Authors:** Octavian Enciu, Elena Adelina Toma, Adrian Tulin, Dragos Eugen Georgescu, Adrian Miron

**Affiliations:** 1Department of Surgery, Carol Davila University of Medicine and Pharmacy, 020021 Bucharest, Romania; octavian.enciu@umfcd.ro (O.E.); gfdragos@yahoo.com (D.E.G.); dramiron@yahoo.com (A.M.); 2General Surgery Department, Elias Emergency University Hospital, 011461 Bucharest, Romania; 3Faculty of Medicine—Discipline of Anatomy, Carol Davila University of Medicine and Pharmacy, 020021 Bucharest, Romania; dr_2lin@yahoo.com; 4I. Cantacuzino Clinical Hospital, 030167 Bucharest, Romania

**Keywords:** situs inversus totalis, laparoscopic cholecystectomy, heuristics, mirror surgery

## Abstract

Background: Laparoscopic cholecystectomy in situs inversus totalis (SIT) is a technically and physically demanding procedure for surgeons and there is still a lack of consensus regarding the best technical approach in such cases. We conducted a systematic review and meta-analysis to evaluate port placement, the dominant hand of the surgeon, preoperative imaging, morbidity, and mortality. Methods: We searched MEDLINE, SCOPUS, Web of Science, and the Cochrane Library for studies of patients with SIT that underwent laparoscopic cholecystectomy. Of 387 identified records, 101 met our inclusion criteria, all of them case reports or case series of maximum of 6 patients. Results: Out of the 121 patients included in the analysis, 94 were operated on using a “mirrored American” technique, 12 using the “Mirrored French”, 9 employed single-port techniques, and 6 described novel port placements. Even though most surgeries were conducted by a right-handed surgeon (93 cases), surgeries performed by the seven left-handed surgeons yielded shorter intervention times (*p* = 0.024). Preoperative imaging (CT, MRI, MRCP, ERCP) also correlated with a lower duration of surgery (*p* = 0.038. Length of stay was associated with the type of disease, but not with other studied endpoints. Morbidity was less than 1%, and conversion rates and mortality were nil. Conclusions: Cholecystectomy in SIT is a safe but challenging procedure and surgeons should prepare in advance for the unfamiliar aspects of completing such a task. While preoperative imaging and a left-handed surgeon are beneficial in terms of surgery length, when these are not available surgeons should focus on achieving the most comfortable setting based on their experience and tailor their approach to the patient at hand. Further studies are needed in order to properly describe and evaluate intraoperative findings as well as surgeon-dependent factors that could improve future recommendations.

## 1. Introduction

Laparoscopic cholecystectomy has been the standard of care for symptomatic cholelithiasis for more than two decades, but patients with situs inversus totalis (transposition of thoracic and abdominal organs to the opposite side of the body) might prove challenging even for experienced surgeons; therefore, thorough preoperative planning is advisable when faced with such cases.

Situs inversus totalis is difficult to evaluate in terms of incidence at birth due to its possible coexistence with other congenital anomalies that are incompatible with life, but Charles W. Mayo reported in 1949 an incidence of roughly 1 in 20,000 admitted patients, of which almost half needed surgery, and seven had diseases of the biliary tract [1]. The first reported laparoscopic procedure in such a patient was in 1991 [2]. Since then, multiple case reports have described a variety of techniques using mirror placement of ports, trying to refer cases to left-handed surgeons, employing single-port surgery, and relying on thorough preoperative imaging, and Chaouch et al. even published a systematic review concerning this issue [3]. Most authors seemed to rely heavily on previously published cases and tried to adapt or improve the reported placement of both surgical teams and operating trocars for the best possible results in terms of heuristics and ergonomics. However, there has been no meta-analysis of these sparse reports in order to try to establish if any particular surgical setting was best in terms of reproducibility and outcomes.

We aim to present an update to already-available reviews (systematic and literature reviews), as well as a meta-analysis of the most important aspects that can help or, on the contrary, be challenging for the surgeons faced with a patient with situs inversus totalis who is a candidate for laparoscopic cholecystectomy. The main outcome sought is whether a recommendation for the standardized technique can be issued to ensure safe completion of the surgical procedure, irrespective of the surgeon’s experience and type of procedure (elective or emergency). The objectives of the analysis are to evaluate whether the single port laparoscopic cholecystectomy (SPLC) might be superior to the classic multiple-port laparoscopic cholecystectomy (LC), if left-handed surgeons are faster in such an unfamiliar setting, whether preoperative imaging (CT, MRI, MRCP, ERCP) improve operative times and reduce complications, and which of the trocar/operator placements yielded the best results in terms of duration of surgery, morbidity, and length of postoperative stay.

## 2. Materials and Methods

This systematic review was conducted based on the Preferred Reporting Items for Systematic Reviews and Meta-Analysis (PRISMA) methodology [4]. All subjects gave their informed consent for inclusion before they participated in the study. The study was conducted in accordance with the Declaration of Helsinki, and the protocol was approved by the Ethics Committee of the Elias Emergency University Hospital–approval no. 8033/2021 [5].

### 2.1. Definition of Situs Inversus Totalis

A patient with situs inversus totalis must have a complete transposition of all thoracic and abdominal organs, unlike patients diagnosed with partial situs inversus, more commonly referred to as levocardia (left-sided heart) or situs ambiguus (also called heterotaxy syndrome), where there are an uncertain number of disturbances of the usual left–right disposition of organs, but cannot be classified as either situs inversus, or situs solitus (normal distribution of organs) [6].

Kartagener’s syndrome is a genetic triad comprised of situs inversus, chronic sinusitis, and bronchiectasis.

Terms sometimes erroneously used to describe patients with abdominal malposition of the gallbladder are sinistroposition and left-sided gallbladder, which actually simply describe the insertion of the gallbladder to the left of the falciform ligament, but with the liver in its usual place in the abdominal cavity.

### 2.2. Research Strategy and Articles Inclusion

A systematic search for all articles regarding laparoscopic cholecystectomy in patients with situs inversus totalis was conducted by two reviewers independently (OE and EAT), in The Cochrane Library, MEDLINE (via PubMed), Scopus, and Web of Science. We used the following medical headings terms (MeSH) and phrases: cholecystectomy; laparoscopic; single-port; situs inversus totalis; Kartagener’s syndrome; left-sided gallbladder; sinistroposition; with the appropriate logical operators (AND, OR, NOT) and employed the “related articles” function where it was available.

Duplicate studies resulting from the search were excluded manually. Appendix A search of Google Scholar and previous reviews’ references for additional articles was also conducted independently. Afterward, some articles were excluded after title screening, followed by exclusion after abstract evaluation. Full-text versions of all remaining articles were read and evaluated and all studies that met the necessary criteria were included in the systematic review.

If the two independent reviewers (OE, EAT) could not reach an agreement regarding the research and inclusion of certain articles, a third reviewer would be called in to reach a consensus (AM).

Articles were considered for inclusion regardless of publication status, sample size, type of disease (chronic or acute cholecystitis, complicated or not), SPLC, or LC, as long as they described at least one of the stated objectives of the meta-analysis. Eligibility criteria for patients were broad, including all reported cases, no matter the sex, ethnicity, and previous medical history, but set a cut-off value of 16 years old for age (patients aged 15 and below were considered pediatric patients). Studies written in languages other than English, that described procedures other than cholecystectomy or other procedures performed simultaneously, studies not reporting at least one outcome of interest or mistakenly reporting sinistroposition as situs inversus, pediatric patients, and robotic surgery patients were excluded.

The final results of the research are outlined in the PRISMA flow diagram below (Figure 1).

### 2.3. Bias Risk Assessment in Results of Included Studies and Results of Meta-Analysis

The risk of bias was assessed using the revised Cochrane ‘Risk of bias’ tool for randomized trials (RoB 2.0), by two of the authors (AT, OE) [7].

Due to the fact that all of the studies included are exclusively case reports or case series, it is unclear what the risk of bias in the results of the respective studies is and a tool to evaluate the methodological quality of these types of articles has not yet been standardized [8]. However, we consider the use of evidence derived from these case reports acceptable in the absence of higher-level evidence in order to formulate recommendations and aid the decision-making process while still maintaining a rigorous evidence-based practice.

We tried to minimize the risk of bias regarding the results of endpoints included in the meta-analyses due to the absence of data from studies that should have been included in the review by having independent researchers extract said data, as well as appoint a third author to check the results before statistical analysis and have a fourth person arbiter lack of consensus between researchers. Considering that restriction of the sample to those patients with complete endpoint data reported would have excluded a significant number of reports and would have increased selection and reporting bias, we collected individual data from each study and chose to summarize missing information for each variable.

### 2.4. Data Extraction and Statistical Analysis

The resulting data was extracted and organized by two authors (OE, EAT) and checked by a third author (DG). In case of disagreements on unclear wording or stated parameters, another author was consulted (AM).

The focus of this meta-analysis was the evaluation of the employed surgical techniques: mirrored American trocar placement, mirrored French trocar placement, and SPLC. Mirrored American LC technique was defined as follows: the patient in supine position, placement of the optical port in the umbilical area (sub- or supra-umbilical), another port in the lower epigastric area, and one or two ports in the left subcostal region (usually on the midclavicular and anterior axillary lines), with the surgeon on the right side of the patient, the first assistant on the left side and the camera person next to the surgeon. Mirrored French LC was defined as: patient in lithotomy position, placement of the optical port in the umbilical region, another port right below the epigastric area, one on the left anterior axillary line, above the iliac spine, and the last one a few centimeters to the right of the median line, between the xiphoid process and the umbilicus. SLPC was defined as: cholecystectomy performed entirely through a single incision.

A secondary point of interest was preoperative advanced imaging (meaning more specific and sensitive in the description of local anatomy than ultrasonography), be it CT, MRI, MRCP, or ERCP.

In conducting this comparison, other endpoints evaluated as secondary outcomes were: the need for additional ports, conversion to LC (for SPLC), conversion to open surgery (for SPLC and LC), duration of the procedure, length of postoperative stay, morbidity (complications or adverse events during hospital stay or after discharge) and mortality. We also noted patient demographics (sex and age), the dominant hand of the surgeon, and the need for intraoperative cholangiography or laparoscopic ultrasonography, rendez-vous ERCP, or laparoscopic common bile duct exploration, which could be expected to prolong the duration of surgery in patients with situs solitus, not just in cases with situs inversus.

For articles where the information was incomplete or unreported, corresponding authors were contacted in order to obtain the data.

Data analysis was conducted using IMB SPSS Statistics V.26.0 (IBM Corporation, Chicago, IL, USA, 2019).

Descriptive statistics were used to present the general characteristics of the resulting samples. Percentages, means, and medians did not include invalid patient data in the denominators (missing data was excluded from individual analysis). Statistical analysis was carried out employing Shapiro–Wilk’s test to assess distribution of continuous variables, the χ^2^ or Fisher’s exact test for categorical variables, and Spearman’s r, Mann–Whitney or Kruskal–Wallis tests for continuous variables (a *p* value equal to or lower than 0.05 was considered statistically significant).

## 3. Results

### 3.1. Case Report

The case that ignited the research was of a 38-year-old female who was admitted to the surgical department of our hospital following multiple episodes of nausea, dizziness, and abdominal pain predominantly in the epigastric region and the left upper quadrant. She underwent an unsuccessful surgery during her childhood for suspected appendicitis, but the surgeon did not manage to find the appendix with a classic McBurney incision (on the right side of the patient).

The clinical evaluation revealed mild pain when palpating the upper half of the abdomen, with no signs of tenderness. Her blood panel was unremarkable, including normal transaminase, lipase, bilirubin, C reactive protein, and procalcitonin levels. An abdominal ultrasound was performed and it demonstrated a distended left-sided gallbladder with slightly thickened walls and multiple small calculi (maximum of 1 cm in diameter), with no other signs indicating obstruction of the biliary tree. An abdominal CT was deemed advisable for appropriate preoperative planning, to observe the exact disposition of the gallbladder and the main biliary duct and exclude the possibility of sinistroposition of the gallbladder. Situs inversus totalis was confirmed (Figure 2).

Having envisaged the difficulty of a mirror operation, extensive research of the literature was employed in order to evaluate trocar positioning and choose the best option for the right-handed surgeon and there were essentially two types of laparoscopy port placements: the “American mirror technique” and the “French mirror technique”. As far as port placement is concerned, we considered that the primary surgeon, being right-handed, would face difficulty when dissecting with the left hand through the epigastric port while the right hand was used for traction on Hartman’s pouch. Therefore, we decided to operate in the lithotomy (“French”) position but employed the “American” trocar positioning, with the surgeon between the patient’s legs, the assistant to the left side of the patient retracting the gallbladder fundus and infundibulum, and the camera operator stood to the right of the patient. A 10 mm optic port was placed in the standard umbilical area, another 10 mm port in the epigastric area 10 cm below the xiphoid process, a third 10 mm port on the midclavicular line 10 cm below the costal margin, and a fourth 5 mm trocar in the left flank on the anterior axillary line. This allowed the surgeon to dissect and retract with both hands, as the situation demanded, and the lower epigastric port created a more efficient setting to avoid excessive stress on the left hand being used for more complex movements in an unfamiliar position (Figure 3). The dispersion of the operating team in this manner allowed for better movement, without concerns regarding the interlocking of hands and/or instruments. The operation was successfully completed within 36 min and the patient’s postoperative course was without complaints. The patient recovered very quickly and was discharged 2 days postoperatively.

Postoperative pathological examination confirmed the presence of gallstones with chronic cholecystitis. No abnormalities were observed at the 30-day follow-up.

### 3.2. Literature Review

The initial database search identified 387 reports, of which 101 met the eligibility criteria for our systematic review. These included 87 case reports (87 patients) and 14 case series of 2 to 6 patients (33 patients), for a total of 121 patients including our own. The studies were published between 1991 and 2021. All included studies can be found in the Appendix A.

The final patient sample that was analyzed was comprised of 85 women and 36 men, with a mean age of 45.68 and 51.86 years, respectively, as detailed in Table 1.

Fifty-eight patients (47.54%) underwent additional high-resolution imaging protocols preoperatively to assess abdominal anatomy: thirty-nine patients underwent abdominal CT scans only, MRI/MRCP scans were used in fourteen cases, and five patients underwent both CT and MRI/MRCP. One patient underwent a HIDA scan (cholescintigraphy) and in another case, a drip-infused cholangiogram was performed preoperatively.

The diagnosis in seventy-three of these patients was chronic cholecystitis or cholelithiasis (60.83%), acute cholecystitis in thirty-two cases (26.6%), while four patients developed acute pancreatitis (3.33%), nine were diagnosed with choledocholithiasis (7.5%) and one was admitted to the hospital for cholangitis (0.83%). The findings are summarized in Table 2.

Endoscopic retrograde cholangiopancreatography (ERCP) was recorded in twelve cases, nine patients underwent the procedure preoperatively (it failed in two cases), two postoperatively and in one case it was successfully completed as a laparo-endoscopic rendezvous procedure.

The dominant hand of the surgeon was recorded in 103 cases: 93 of the surgeons were right-handed (90.29%), seven were left-handed (6.79%), two were ambidextrous (1.94%) and one case was operated on by two surgeons, one right-handed and one left-handed, alternating their roles as main operator [9].

As far as the operative techniques employed, 110 reports described a four-port technique, while a three-port cholecystectomy was performed in five cases: 94 surgeons used the mirrored version of the American port placement–77.68% and 12 cases or 10% of surgeons preferred the mirrored French technique. Nine patients underwent single-port surgery (7.5%), while six patients were operated on using novel port placements, aiming for potentially better ergonomics in approaching this challenging set-up.

Intraoperative cholangiography was recorded in 12 cases (9.91%), in order to either confirm the diagnosis of choledocholithiasis, or to clarify the anatomy of the biliary tree and, additionally, indocyanine green (ICG) fluorescent cholangiography was used in two other cases. Two patients needed laparoscopic common bile duct exploration (LCBDE) and one underwent a laparoscopic choledocoduodenostomy due to significant dilation of the CBD [10].

The duration of surgery was recorded in 78 cases and resulted in a mean value of 71.21 min, while postoperative hospital stay was recorded in 105 cases, with a mean of 2.05 days. The data is summarized in Table 3. Detailed data on duration and postoperative stay depending on surgical technique and the dominant hand of the surgeon are presented in Table 4 and Table 5.

Statistical analysis showed a significant correlation between the dominant hand of the surgeon and the duration of the surgery, with a faster outcome for left-handed surgeons (*p* = 0.024), and preoperative imaging (CT, MRI, MRCP, ERCP) was also beneficial in reducing operative time (*p* = 0.038). As for the postoperative stay, the only variable that was significant statistically was the disease that led to surgery (*p* < 0.001). Intraoperative cholangiography was not statistically significant in relation to either endpoint. These findings are summarized in Table 6 and Figure 2a,b.

Morbidity was 0.84%, while mortality and conversion rates were nil.

## 4. Discussion

Laparoscopic cholecystectomy is the standard of care for gallstone disease, and may be one of the most standardized and studied procedures in laparoscopic surgery. This is partially due to its status as one of the most frequent procedures performed in general surgery and likely the first laparoscopic procedure a general surgeon learns during their residency.

Paradoxically, the learning curve is yet to be defined, as the rate of bile duct injury declines after only 50 procedures, while the operative time still decreases after 200 procedures [11]. This only leads to the question at hand—how prepared can you be when faced with a mirror situation? Because we are most likely not habitually mirror writers and readers like Leonardo DaVinci and once you are set in the paradigm of a routine intervention it is very hard to get out of the box [12]. The learning curve is usually defined in a retrospective fashion, rather than based on prospectively acquired sets of data. Hamdorf and Hall provide an insightful description of, “the relationship between learning styles and surgical decision making, and it is appreciated that there are many different styles of learning: concrete experience, abstract conceptualization, active experimentation and reflective observation”, concluding that while it may be feasible to evaluate certain aspects of surgery, cognitive skills are just as much a part of this curve and a lot harder to appraise [13]. Moreover, there is a lack of standardization in defining which endpoints actually measure competence and surgical safety and until such definitions arise, it is unlikely there will be an agreed-upon number of cholecystectomies from whereon one can say they are proficient [14].

Laparoscopic surgery implies that the information is acquired through visual perception and is processed in a heuristic manner. Visual perception is not equivalent to reality but is more of a mental construct in which the continuous stream of information is processed based on expectations, previous experiences, and memories. The most influential rule for avoiding bile duct injury is “the critical view of safety” stated by Strasberg but during laparoscopic cholecystectomy in SIT we found ourselves looking through Calot’s triangle from lateral to medial–exactly the opposite as we would have during a routine intervention on the right side, because that is what we were accustomed to [15]. Moreover, since the first reported cholecystectomy in an SIT patient in 1991, authors have started to consider the stress on both the body and mind of the surgeon as issues of interest, on par with the actual technical aspects of the surgery [16,17,18]. The interlocking of instruments, hyperflexion of the surgeon’s trunk and left hand, subsequent early fatigue, and place-changing mid-surgery, all while trying to re-train the brain to a new, unfamiliar view, is extremely challenging.

Beyond this, surgeons need comfort during surgery. Even though most interventions surveyed in our systematic review had a reasonable operative time, the comfort of the surgeon is not discussed in detail. We can presume that in most of the published cases the operating surgeon has considerable experience and dexterity (and many authors do mention this and underline it even) because one can imagine the ordeal it is for a beginner to operate in a mirrored fashion. The approach we have chosen after researching the relevant literature was the most straightforward and comfortable for our operating team which benefitted from oversight from the most senior surgeons in the clinic, as well as the advantage of having operated together on numerous occasions. While it does not resemble any conventional trocar positioning used for laparoscopic cholecystectomy, but more the approach for hiatal hernia, maybe this is the key for unconventional laparoscopic surgery–just find the best triangulation.

One of the arguments for proficiency in this setting is that laparoscopic cholecystectomy in patients with SIT is the revenge of nature in order to understand the importance of being left-handed [19]. More than 90% of the general population are right-handed or prefer to use predominantly the right hand [20]. This leads to difficulties in training as instruments are designed and optimized for right-handed manipulation and mentorship is difficult when working in tandem with a right-handed surgeon [21]. Even though right-handed individuals are reported to have a lower error rate when performing endoscopic tasks, the differences reflect their innate abilities, not their potential to learn and improve [22]. As left-handed surgeons are underrepresented, hoping for referral to a left-handed surgeon for laparoscopic cholecystectomy is unrealistic. Additionally, they themselves face cognitive dissonance regardless of their perceived advantage, being mentally prepared for a right-sided gallbladder just as right-handed surgeons are. In our survey, out of seven interventions carried out by left-handed surgeons, only four communicated near-complete data and reported shorter operative times. (Figure 4b) They all used the mirror American (MirA) trocar positioning. Considering the small sample size and missing data regarding the technical difficulty of the intervention, a definitive conclusion cannot be drawn.

Regarding patient demographics, gallstone disease tends to display the same prevalence as in the normal population, affecting women more frequently [23]. In our survey, the youngest patient operated on was 16 years of age while the oldest was 88, with a mean of 48 years (Table 1).

No conversion to open surgery was reported and taking into consideration missing data, in only one case of acute cholecystitis the authors reported morbidity in the form of residual common bile duct stones that was successfully managed by postoperative ERCP 20 days after surgery [24]. However, it should be specified that most of the case reports included in the study stated that the surgeon was an experienced one, and as previously mentioned, the rate of bile duct injury decreases after the completion of a high number of cholecystectomies. The duration of surgery for cholecystectomy alone was fairly standard with a mean duration under 100 min (range 30–150 min) while for additional maneuvers required for choledocholithiasis the mean operative time was 112.33 (range 50–240) (Table 2). The mean duration of surgery taking all reported cases into consideration was 71.21 min while the mean postoperative stay was 2.05 days (range 1–11) and surgery length did not correlate with the length of stay (r = 0.032, *p* = 0.786) (Table 3). The postoperative stay was dependent only on the type of disease (*p* = 0.001) which is in line with large-scale studies conducted on situs solitus patients and is to be expected to be shorter for patients with uncomplicated lithiasis as opposed to those who develop acute cholecystitis, choledochal lithiasis, pancreatitis or cholangitis.

The duration of surgery was shorter in patients that had additional high-resolution preoperative imaging (*p* = 0.038, Figure 2a). All patients had preoperative ultrasounds–the gold standard for diagnosing gallbladder calculi, but we reckon that in cases where more advanced imaging techniques are not available or not used, especially in an emergency setting or in cases of scleroatrophic gallbladders, intraoperative ultrasound may be extremely helpful in clarifying local anatomy [25,26]. The median duration of surgery was shortest for the MirF technique (60 min) while the longest was recorded for the single-port techniques (75 min).

Adding more detail to the operative time analysis, like putting out a fire with gasoline, we find that the left-handed surgeons operated faster (*p* = 0.024). If we were to combine advanced preoperative imaging with a left-handed surgeon operating in a MirF port placement, would we obtain the fastest and most comfortable laparoscopic cholecystectomy in SIT? This hardly reflects reality—the fact is that most surgeons are right-handed and operate in an MirA fashion, and probably these are the most in need of information that can be forced out of this systematic review.

Patle et al. have the largest series of patients with SIT that have undergone laparoscopic cholecystectomy, and this group aimed to advance their approach in the MirA fashion by placing the operator between the legs of the patient in a lithotomy position as they suggested, but placed the trocars lower and used the midclavicular port for dissection as well [27]. There were 17 mentions of using the midclavicular port for dissection, but in one case it was abandoned due to interference with the instruments handled with the left hand, so this artifice is not universally applicable either [28]. We also identified six cases that described entirely new dispositions of trocars: two working ports in the iliac fossae, two working ports on the midline, two cases where the surgeon and first assistant on the left side of the patient dissecting through a port situated on the posterior axillary line, as well as other two that had the surgeon on the right side with the patient in lithotomy position and port placements different than the three “standard” techniques, described [29,30,31,32,33]. However, these cases did not result in an improved duration of surgery or hospital stay.

SIT is a challenging aspect for surgeons performing any number of laparoscopic procedures, not just cholecystectomies. Akbulut et al. published a review of appendectomies in SIT patients in 2010, and out of 95 cases, only 8 underwent laparoscopic resections [34]. Colorectal procedures (for both benign and malignant ailments) have also been reported with favorable outcomes, but authors still underline the need for diligent preoperative planning, including but not limited to CT angiography, adapted trocar placement, and careful dissection [35,36,37]. Meanwhile, gastric interventions (either for perforated ulcers, gastric cancer, or bariatric procedures) seem to be undertaken as a somewhat easier endeavor, arguably due to the similar alignment of the stomach relative to the midline, with some reports mentioning no change in patient, operator, or trocar placement in such cases [18,38,39].

Future studies could benefit from improvements in reporting outcomes as underlined by O’Connor et al. who proposed a minimum synoptic operation template for cholecystectomies, or by using an operative difficulty grading scale such as the one described by Nassar et al. [40,41]. This would be beneficial when conducting such systematic reviews and meta-analyses, considering the low probability of future randomized controlled trials due to the scarcity of patients with SIT and the fact that the duration of surgery and postoperative stay could not tell the full story when one considers the actual discomfort and struggles of the operating team.

## 5. Conclusions

Laparoscopic cholecystectomy is not comfortable in the setting of situs inversus totalis but it is safe and maintains its gold standard etiquette. The operating surgeon must be at ease with the trocar placement, and in our opinion should aim for the best triangulation, which can be achieved either by employing mirrored positioning of the usual ports or by adapting techniques that they are already comfortable with. A surgeon’s dexterity and training, preoperative imaging, and careful planning might successfully substitute a left-handed operator. Future case series and case reports should include further analysis and description of quantifiable items related to difficulty in operating on such patients.

## Figures and Tables

**Figure 1 diagnostics-12-01265-f001:**
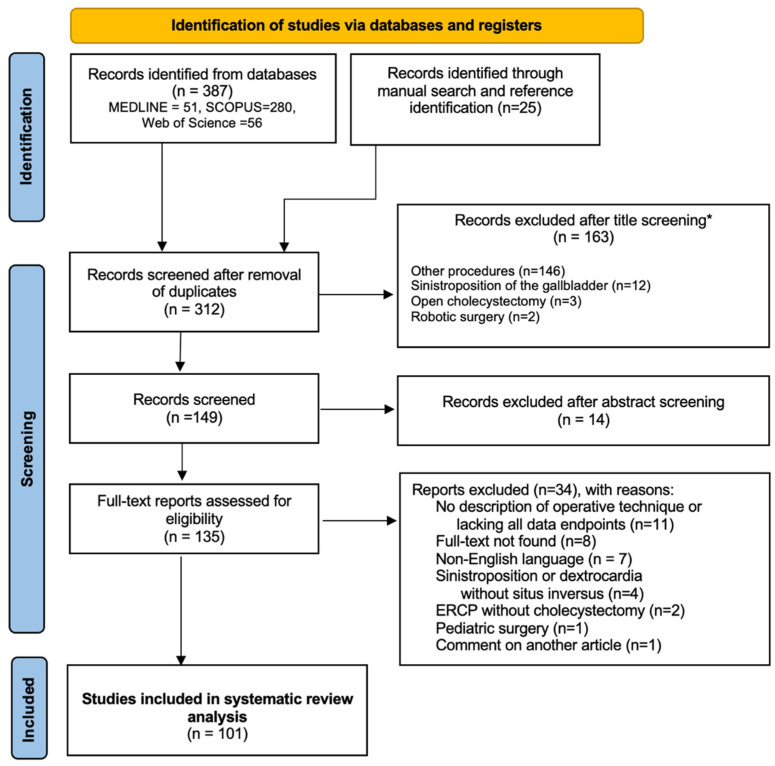
PRISMA flow diagram for research results. * all items excluded manually.

**Figure 2 diagnostics-12-01265-f002:**
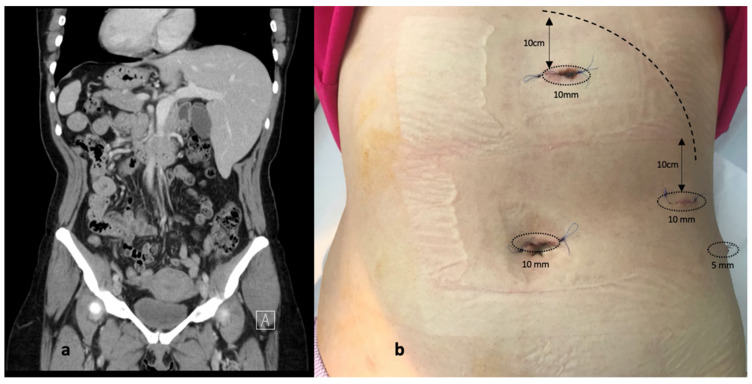
(**a**) Abdominal CT scan demonstrating situs inversus totalis (**b**) Disposition of ports (postoperative aspect).

**Figure 3 diagnostics-12-01265-f003:**
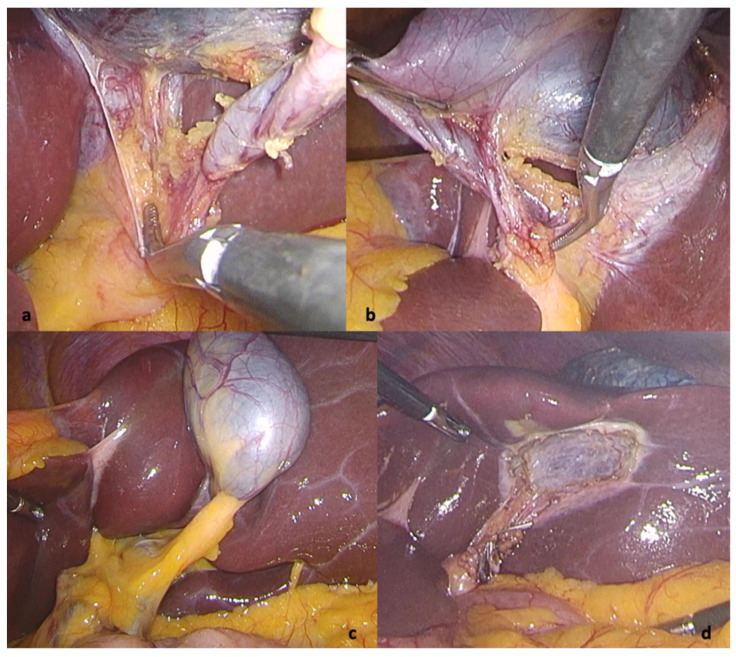
Intraoperative view: (**a**) critical view of safety (**b**) lateral to medial view of the cystic duct (**c**) view of the Calot region at the beginning of surgery (**d**) view after successful cholecystectomy.

**Figure 4 diagnostics-12-01265-f004:**
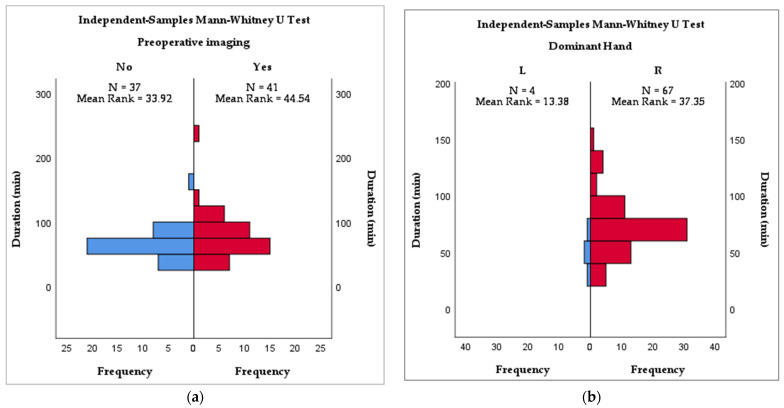
(**a**) Mann–Whitney analysis of preoperative imaging and duration of surgery (**b**) Mann–Whitney analysis of dominant hand of the surgeon and duration of surgery.

**Table 1 diagnostics-12-01265-t001:** Patient demographics.

Variable	Sex	N	Mean	Minimum	Maximum
Age	F	85	45.68	16	79
	M	36	51.86	21	88
	Total	121	47.52	16	88

**Table 2 diagnostics-12-01265-t002:** Duration of surgery, postoperative stay, and morbidity according to disease.

Disease		N	Mean	Std. Dev.
**N/A**	Duration (min)	2	100.00	70.711
(N = 2)	Postop Stay (days)	2	4.50	3.536
	Morbidity	0		
**AC**	Duration (min)	16	67.06	23.510
(N = 32)	Postop Stay (days)	28	2.82	2.127
	Morbidity	31		
**CC**	Duration (min)	50	68.14	21.089
(N = 73)	Postop Stay (days)	63	1.52	0.981
	Morbidity	73		
**CHL**	Duration (min)	6	112.33	67.331
(N = 9)	Postop Stay (days)	8	2.88	2.475
	Morbidity	9		
**CHO**	Duration (min)	0		
(N = 1)	Postop Stay (days)	1	2.00	
	Morbidity	1		
**AP**	Duration (min)	4	50.00	18.257
(N = 4)	Postop Stay (days)	3	2.00	1.732
	Morbidity	4		

N/A—not available; AC—acute cholecystits; CC—chronic cholecystitis; CHL—choledocolithiasis; CHO—cholangitis; AP—acute biliary pancreatitis.

**Table 3 diagnostics-12-01265-t003:** Summary of surgery duration and postoperative hospital stay.

Variable	N	Min	Median	Max	Spearman’s r
Duration (min)	78	30	68	240	r = 0.032 *p* = 0.786
PO Stay (days)	105	1	1	11	

**Table 4 diagnostics-12-01265-t004:** Surgery duration, morbidity, and postoperative stay sorted by operative technique.

Operative Technique			Duration (min)	Morbidity	Postop Stay (Days)
**Other Techniques**	N = 6	Valid	4	6	6
		Missing	2	0	0
	Median		72.00	0.00	1.00
	Percentiles	25	40.00	0.00	1.00
		50	72.00	0.00	1.00
		75	74.75	0.00	3.50
**MirA**	N = 94	Valid	60	93	82
		Missing	34	1	12
	Median		65.00	0.00	1.00
	Percentiles	25	50.00	0.00	1.00
		50	65.00	0.00	1.00
		75	80.00	0.00	3.00
**MirF**	N = 12	Valid	7	10	8
		Missing	5	2	4
	Median		60.00	0.00	2.50
	Percentiles	25	55.00	0.00	2.00
		50	60.00	0.00	2.50
		75	88.00	0.00	4.50
**SP**	N = 9	Valid	7	9	9
		Missing	2	0	0
	Median		75.00	0.00	2.00
	Percentiles	25	54.00	0.00	1.00
		50	75.00	0.00	2.00
		75	90.00	0.00	2.00

MirA—mirrored American; MirF—mirrored French; SP—single-port.

**Table 5 diagnostics-12-01265-t005:** Surgery duration, morbidity, and postoperative stay sorted by dominant hand of the surgeon.

Dominant Hand				Duration (Min)	Morbidity	Postop Stay (Days)
**N/A** (N = 18)		N	Valid	6	18	15
			Missing	12	0	3
		Median		67.50	0.00	2.00
		Percentiles	25	45.00	0.00	1.00
			50	67.50	0.00	2.00
			75	127.50	0.00	3.00
**AMBI** (N = 3)		N	Valid	1	3	2
			Missing	2	0	1
		Median		106.00	0.00	1.00
		Percentiles	25	106.00	0.00	1.00
			50	106.00	0.00	1.00
			75	106.00	0.00	1.00
**L** (N = 7)		N	Valid	4	7	5
			Missing	3	0	2
		Median		47.50	0.00	1.00
		Percentiles	25	33.75	0.00	1.00
			50	47.50	0.00	1.00
			75	57.50	0.00	7.00
**R** (N = 93)		N	Valid	67	90	83
			Missing	26	3	10
		Median		70.00	0.00	1.00
		Percentiles	25	55.00	0.00	1.00
			50	70.00	0.00	1.00
			75	80.00	0.00	2.00

N/A—not available; AMBI—ambidextrous; L—left-handed; R—right-handed.

**Table 6 diagnostics-12-01265-t006:** Association between duration of surgery/postoperative stay and endpoints.

	Variable	95% CI for ρ	*p*-Value
**Duration of surgery (min)**	Disease	(−0.311, 0.136)	0.068
	Preoperative imaging	(0.007, 0.430)	0.038
	Dominant Hand	(−0.000, 0.442)	0.024
	Operating Technique	(−0.190, 0.267)	0.731
	IOC	(−0.045, 0.394)	0.114
**Postop Stay (days)**	Disease	(−0.525, −0.189)	0.001
	Preoperative imaging	(−0.164, 0.219)	0.771
	Dominant Hand	(−0.268, 0.144)	0.543
	Operating Technique	(−0.175, 0.219)	0.079
	IOC	(−0.325, 0.061)	0.174

## Data Availability

Not applicable.

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
