# Peer review of "Look beyond the Mirror: Laparoscopic Cholecystectomy in Situs Inversus Totalis—A Systematic Review and Meta-Analysis (and Report of New Technique)"

_diagnostics, 2022, doi:10.3390/diagnostics12051265_

Round 1

Reviewer 1 Report

well done and interesting paper to read and to know about

Author Response

Thank you very much for your comment and appreciation!

Reviewer 2 Report

This is a well written, sound review of a particular aspect of general surgery. I have no comments on it

Author Response

Thank you for your praise and your review!

Reviewer 3 Report

Dear Editor, 

 This is an interesting review of laparoscopic surgery for situs inversus totalis. The updated information provides current knowledge for this rare situation based on case repots and literature review. The followings are my comments. 

  1. Table 2,3,4 and 5  are too long to read. Please choose either mean or median depends on the data distribution ( normal distribution or not normal distribution ).
  2.  Table 6 , express p with  95% CI  is enough 
  3. Line 288, Paradoxically, the learning curve is yet to be defined, as the rate of bile duct injury  . While discussion the larning curve, do authors have data to support this view in this analysis ?
  4. I would like to read more discussion about surgical aspects in this rare situation rather than surgical aspects for laparoscopic cholecystectomy. The authors may cite some reference in other field of surgery.

Author Response

  1. We have made the suggested changes to the Tables, thank you for your input.
  2. We had only included p and CI, we could not make any other changes to the 6th Table.
  3. We have cited additional data regarding the learning curve, as well as how it related to the cases included in our review. 
  4. Due to the particular disposition of the gallbladder in the human body, as well as the apparently higher reported incidence of cholecystitis in patients with SIT,  when compared to other procedures referenced in the literature (colorectal, gastric), we chose to emphasize the adjusments the surgeon needs to make to increase the odds of a favorable outcome. However, we did add a paragraph mentioning other laparoscopic procedures that were carried out successfully in SIT patients - and found it interesting that due to the relationship of organs such as the rectum and the stomach to the midline, they might have even kept the standard trocar and operator placement. 

Thank you very much for your valuable input, we hope you find the changes we made satisfactory.

Reviewer 4 Report

THIS IS AN INTERESTING PAPER

i have some minor criticisms:

-about the learning curve: please add more iformation about the learning curve if it is present in literature a difference between left or right hand surgeon.

i think that the problem is not the hand but the surgeon!

so for this reason, it should be discuss into the discussion

- again, learning curve is very difficult to express , so i suggest to better explain what do you mean with long time results and learning curve

-

Author Response

We cited additional information regarding the learning curve, as well as some more information about left-handed vs right-handed surgeons, but the literature is scarce when it comes to this subject. 

We do hope you are pleased with the changes we made and find the explanations we found do shed more light on this matter. 

Round 2

Reviewer 3 Report

Dear Editor

 The authors responded to the questions adequately. I have no further questions.